



**Evaluating the response of $\delta^{13}$C in *Haloxylon ammodendron,* a**
**dominant C$_4$ species in Asian desert ecosystem, to water and nitrogen**
**addition as well as the availability of its $\delta^{13}$C as the indicator of water**
**use-efficiency**
**Zixun Chen[1,2], Xuejun Liu[2,3], Xiaoqing Cui[2,3], Yaowen Han[2], Guoan Wang[1,2]\*, Jiazhu Li[4]\***
1. Key Lab of Plant–Soil Interaction, College of Resources and Environmental Sciences, China
Agricultural University, Beijing, 100193, China.
2. Beijing Key Laboratory of Farmland Soil Pollution Prevention and Remediation, Department of
Environmental Sciences and Engineering, College of Resources and Environmental Sciences,
China Agricultural University, Beijing, 100193, China.
3. Xinjiang Institute of Ecology and Geography, Chinese Academy of Sciences, Urumqi, 83011,
China
4. Institute of Desertification Studies, Chinese Academy of Forestry, Beijing, 100192, China.
**\*Corresponding author:**
**Guoan Wang, gawang@cau.edu.cn**
**Jiazhu Li, leejzids@caf.ac.cn**



## Abstract

Variations in precipitation and atmospheric N deposition affect water and N
availability in desert, and thus may have significant effects on desert ecosystems.
*Haloxylon ammodendron* is a dominant plant in Asian desert, and addressing its
physiological acclimatization to the changes in precipitation and N deposition can
provide an insight into how desert plants adapt extreme environment by physiological
adjustment. Carbon isotope ratio ($\delta^{13}$C) in plants has been suggested as a sensitive
long-term indicator of physiological acclimatization. Therefore, this study evaluated
the effect of precipitation change and increasing atmospheric N depositon on $\delta^{13}$C of
*H. ammodendron*. Furthermore, *Haloxylon ammodendron* is a C$_4$ plant, whether its
$\delta^{13}$C can indicate water use-efficiency (WUE) has not been addressed. In the present
study, we designed a field experiment with a completely randomized factorial
combination of N and water, and measured $\delta^{13}$C, gas exchange and WUE of the
assimilating branches of *H. ammodendron*. $\delta^{13}$C in *H. ammodendron* remained stable
under N and water supply, while N addition, water addition and their interaction
affected gas exchange and WUE in *H. ammodendron*. In addition, $\delta^{13}$C had no
correlation with WUE. This result are associated with the irrelevance between $\delta^{13}$C
and $c_i/c_a$, which might be caused by a special value (0.37) of the degree of
bundle-sheath leakiness ($\varphi$) or a lower activity of carbonic anhydrase (CA) of *H.*
*ammodendron*. Thus, $\delta^{13}$C of *H. ammodendron* cannot be used for indicating its
WUE.



## 1 Introduction

Recently, global precipitation pattern has changed significantly (Frank et al., 2015;

Knapp et al., 2015), and atmospheric N deposition has continued to rise (Galloway et

al., 2004; Liu et al., 2013; Song et al., 2017). Previous researchers have suggested that

arid ecosystems are most sensitive to climate change (Reynolds et al., 2007; Huang et

al., 2016), while global change in precipitation and atmospheric N deposition has an

important impact on water and N availability in desert (Huang et al., 2018). Thus,

these changes may have significant effects on desert ecosystems. *Haloxylon*

*ammodendron* is a dominant species in desert regions, especially in Asia. Studying the

physiological responses of *H. ammodendron* to global change can provide an insight

into how desert plants adapt extreme environment by physiological adjustment.

Carbon isotope ratio ($\delta^{13}C$) in plants depends on the ratio of intercellular to ambient

$CO_2$ concentration ($c_i/c_a$), which reflects the balance between inward $CO_2$ diffusion

rate, regulated by stomatal conductance ($g_s$), and $CO_2$ assimilating rate (A) (Farquhar

and Richards, 1984), and has been suggested as a sensitive long-term indicator of

physiological acclimatization (Battipaglia et al., 2013; Cernusak et al., 2013; Tranan

and Schubertt, 2016; Wang and Feng, 2012). Therefore, investigating the variations in

$\delta^{13}C$ of *H. ammodendron* under water and nitrogen addition can enhance our

understanding of physiological responses of desert plants to future changes in

precipitation and atmospheric N deposition.

A large quantity of works have been devoted to the relationships between $C_3$ plant

$\delta^{13}C$ and water availability or precipitation (e.g., Diefendorf et al., 2010; Kohn, 2010;



Liu et al., 2005; Ma et al., 2012; Serret et al., 2018; Stewart et al., 1995; Wang et al.,
2005, 2008) and nitrogen availability (e.g. Cernusak et al., 2007; Li et al., 2016;
Sparks and Ehleringer, 1997; Yao et al., 2011; Zhang et al., 2015). However, a
relatively small amount of research has focused on the responses of $C_4$ plant $\delta^{13}C$ to
water availability or precipitation (Ellsworth et al., 2017; Liu et al., 2005; Rao et al.,
2017; Wang et al., 2006) and nitrogen availability (Ma et al., 2016; Schmidt et al.,
1993). For $C_4$ plants, $\delta^{13}C$ is controlled by both the $c_i/c_a$ ratio and the degree of
bundle-sheath leakiness ($\varphi$), the proportion of $CO_2$ produced within bundle sheath
cells from $C_4$ acids that leaks back to mesophyll cells (Ellsworth and Cousins, 2016;
Ellsworth et al., 2017; Farquhar, 1983). Thus, the responses of $C_4$ plant $\delta^{13}C$ to water
and N availability are also affected by $\varphi$. Genetic factors control $\varphi$ values, which
causes the interspecific differences in $\delta^{13}C$, even the responses of plant $\delta^{13}C$ to water
and N availability (Gresset et al., 2014). On the other hand, enzymatic activity of
carbonic anhydrase (CA) may influence $\delta^{13}C$ in $C_4$ plants as CA activity is low
(Cousins et al., 2006). CA is an enzyme that catalyzes the hydration of $CO_2$ in
mesophyll cells to form bicarbonate ($HCO_3^-$). Previous studies showed that CA
activity in most $C_4$ plants is usually low, just sufficient to support photosynthesis
(Cousins et al., 2006; Gillon and Yakir, 2000, 2001; Hatch and Burnell, 1990). *H.*
*ammodendron* is a typical $C_4$ plant. How its $\delta^{13}C$ responds to water and N availability
has never been addressed.
Foliar $\delta^{13}C$ in $C_3$ plants has been considered as a useful indicator of intrinsic water
use-efficiency (WUE) (Farquhar, 1983). However, although some studies suggested



that $\delta^{13}C$ of $C_4$ plants could also indicate its WUE (Henderson et al., 1992; Wang et al.,
2005; Cernusak et al., 2013; Ellsworth and Cousins, 2016), this statement is still
controversial. The relationship between $\delta^{13}C$ and WUE is based on the links between
$c_i/c_a$ ratio and $\delta^{13}C$ and between $c_i/c_a$ ratio and WUE (Ehleringer and Cerling, 1995).
For $C_3$ plants, $\delta^{13}C$ always increases with an increase in $c_i/c_a$ ratio; but for $C_4$ plants,
the correlation between $\delta^{13}C$ and $c_i/c_a$ ratio depends on φ value (Cernusak et al., 2013)
and CA activity (Cousins et al., 2006). As mentioned above, φ value is under genetic
control, thus, the correlation between $\delta^{13}C$ and $c_i/c_a$ ratio, as well as the relationship
between WUE and $\delta^{13}C$, shows interspecific difference. Whether $\delta^{13}C$ of *H.*
*ammodendron* indicates WUE has never been evaluated.
In this study, we designed an experiment with multiple water and nitrogen supply in
the southern Gurbantunggut Desert in Xinjiang Uygur Autonomous Region, China.
We measured the $\delta^{13}C$, gas exchange and WUE of the assimilating branches of *H.*
*ammodendron*. We had two objectives. One objective was to evaluate the response of
the dominant plant of Asian desert to future changes in precipitation and atmospheric
N deposition by revealing the effects of water and N supply on $\delta^{13}C$ of *H.*
*ammodendron*. The other was to explore the availability of $\delta^{13}C$ as the indicator of
water use-efficiency in *H. ammodendron*.

**2 Materials and methods**
**2.1 Definitions and Basic Equations**
Stable carbon isotopic ratio ($\delta^{13}C$) of natural materials is expressed as:





$$\delta^{13}C(‰)= \left[ \frac{(^{13}C/^{12}C)_{sample}}{(^{13}C/^{12}C)_{standard}} -1 \right] \times 1000 \qquad (1)$$

where the standard is the carbon dioxide obtained from the Peedee belemnite (PDB)
limestone (Craig, 1957). Farquhar (1983) proposed the pattern of carbon isotopic
discrimination ($\Delta$) in $C_4$ plant:
$$\Delta = \frac{\delta^{13}C_{air} - \delta^{13}C_{plant}}{1+\delta^{13}C_{plant}/1000} \approx \delta^{13}C_{air} - \delta^{13}C_{plant} = a + \left[ b_4 + \varphi \left( b-s \right) - a \right] \frac{c_i}{c_a} \qquad (2)$$

where $\delta^{13}C_{plant}$ and $\delta^{13}C_{air}$ are the $\delta^{13}C$ values of plants and $CO_2$ in the ambient air.
The parameter a (= 4.4‰, Craig, 1954) is the carbon isotopic fractionation in the
diffusion of $CO_2$ into internal leaves; $b_4$ (= -5.9‰, O'Leary, 1984) is the combined
carbon isotopic fractionations occurring in the processes of gaseous $CO_2$ dissolution,
hydration/dehydration reactions of $CO_2$ and $HCO_3^-$ in mesophyll cells, and
$HCO_3^-$ carboxylation by PEP (phosphoenolpyruvate) carboxylase; s (= 1.8‰, O'Leary,
1984) is the carbon isotopic fractionation during diffusion of $CO_2$ out of the
bundle-sheath cells, and b (= 27‰, Farquhar and Richards, 1984) is the carbon
isotopic fractionation of $CO_2$ carboxylation by RuBP (ribulose-1,5-bisphosphate)
carboxylase. The variable $\varphi$ is the proportion of $CO_2$ producing within bundle sheath
cells from $C_4$ acids that leaks back to mesophyll cells, and $c_i/c_a$ is the ratio of
intercellular to ambient $CO_2$ concentration.
Water use-efficiency (WUE) is defined as the amount of assimilated carbon dioxide
by plants under the consumption of per unit water. There are two characteristics of
WUE, instantaneous WUE (ins-WUE) and intrinsic WUE (int-WUE), respectively.
ins-WUE can be calculated by:
$$\text{ins-WUE} = A/E = (c_a - c_i)/1.6v = c_a(1-c_i/c_a)/1.6v \qquad (3)$$



where A is photosynthetic rate, E is transpiration rate and v is calculated
by:
$$v = (e_i-e_a)/p \tag{4}$$
where $e_i$ and $e_a$ are the water vapor pressure inside and outside the leaves, p is the
atmospheric pressure.
The definition of int-WUE is:
$$\text{int-WUE} = A/g_s = (c_a–c_i)/1.6 = c_a(1-c_i/c_a)/1.6 \tag{5}$$
where $g_s$ is stomatal conductance.
**2.2 Study site**
This experiment was conducted at the Fukang Station of Desert Ecology, Chinese
Academy of Sciences, on the southern edge of the Gurbantunggut Desert (44 °26′ N,
87 °54′ E) in northwestern China. The altitude of the study site is 436.8 m above
average sea level (a.s.l.). It is a typical continental arid, temperate climate, with a hot
summer and cold winter in the area. The mean annual temperature is 7.1 °C and the
mean annual precipitation is 215.6 mm, with a potential evaporation of about 2000
mm. The soil type is grey desert soils (Chinese classification) with aeolian sands on
the surface (0-100 cm). The percentages of clay (< 0.005 mm), silt (0.005-0.063 mm),
fine sand (0.063-0.25 mm) and medium sand (0.25-0.5 mm) range from 1.63-1.76%,
13.79-14.15%, 55.91-56.21% and 20.65-23.23%, respectively (Chen et al., 2007). The
soil is highly alkaline (pH = 9.55 ± 0.14) with low fertility. The vegetation is
dominated by *Haloxylon ammodendron* and *Haloxylon persicum* with about 30%
coverage. Herbs include ephemerals, annuals and small perennials, with a cover of ca.





40% (Fan et al., 2013). Although the coverage of the two *Haloxylon* species is a little
lower than that of herbs, the biomass of the former is much larger than that of the
latter, because *Haloxylon* plants are shrubs with an average height of 1.5 m whereas
the latter are very low herbaceous plants. Biological soil crusts are distributed widely
on the soil between the herbs and *Haloxylon*, with almost 40% coverage (Zhang et al.,

160  2007).

**2.3 Experimental design**

A field experiment with a completely randomized factorial combination of water and
nitrogen has been conducted from 2014 to 2017. We designed two water addition
levels (0, 60mm $yr^{-1}$; W0, W1), since precipitation is predicted to increase by 30% in
northern China in the next 30 years (Liu et al., 2010), and three levels of N addition (0,
30, 60 kg N $ha^{-1}$ $yr^{-1}$; N0, N1 and N2), because N deposition has reached 35.4 kg
N $ha^{-1}$ $yr^{-1}$ in the nearby city, Urumqi (Cui et al., 2017) and will double by 2050
relative to the early 1990s (Galloway et al., 2008). Therefore, there were six
treatments (W0N0, W0N1, W0N2, W1N0, W1N1, W1N2) in this experiment. Four
replicates of each treatment were set, making a total of 24 plots with a size of 10 m
×10 m. A small sub-plot with a size of 1.5 m×1.5 m was set in each plot. A
well-grown *H. ammodendron* was enclosed in the center of the sub-plot. The average
height and coverage of an individual *H. ammodendron* were 1.5 m and 1.9 $m^2$,
respectively, and did not vary significantly across the plots. To simulate natural water
and N inputs, the treatments were applied in equal amounts, twelve times, once a
week in April, July and September, as 5 mm $m^{-2}$ of water and 2.5 or 5 kg N $ha^{-1}$ each





week (Cui et al., 2017). Usually, water addition was with a sprinkler kettle, irrigating
over the canopy of *H. ammodendron*.

**179     2.4 Measurements of gas exchange traits and WUE**

Gas exchange traits, including photosynthetic rate (A), stomatal conductance ($g_s$),
transpiration rate (E) and $c_i/c_a$, on the assimilating branches of the *H. ammodendron*
grown in the sub-plots were determined by LI-6400 portable photosynthesis system
on 27-29, June 2016. Then we calculated ins-WUE by the Eq. (3), and int-WUE by
Eq. (5).
At each plot, the top assimilating branches of a mature individual was selected
randomly for the measurement. About 5s was needed for stability after the
assimilating branches was inserted in the cuvette and then the assimilating branches
were measured. We repeated 10 times on the same assimilating branches for each
measurement. We measured gas exchange with a standard 450 mmol $mol^{-1}$ $CO_2$
concentration at a flow rate of 500 mmol $s^{-1}$ above saturation in photo flux density of
1000 mmol $m^{-2}$ $s^{-1}$. Leaf temperature kept stable and varied within 1.0 ℃ during each
measurement.

**193     2.5 Samples collection**

Samples were collected in July 2017. Previous researches have proved that leaf, as the
assimilating organ in plants, was most effective for the assessment of plant $\delta^{13}C$
(Saranga et al., 1999). However, extreme drought in desert ecosystems causes the
degeneration of leave in *H. ammodendron*. Thus we had to collect the assimilating
branches of *H. ammodendron* as our samples, which was its prime assimilating organ.





All *H. ammodendron* individuals grown in plots (10 m × 10 m) were sampled. Eight
pieces of assimilating branches were collected from each individual, two pieces of
assimilating branches were collected at each of the four cardinal directions from the
positions of full irradiance. All assimilating branches from the same plot were
combined into one sample. All plant samples were air-dried in the field and then in
the laboratory. Then the samples were ground into a fine powder using a steel ball
mixer mill MM200 (Retsch GmbH, Haan, Germany) for the measurements of $\delta^{13}$C, N
contents and chlorophyll contents.
**2.6 Measurements of plant $\delta^{13}$C, plant N and chlorophyll contents**
The $\delta^{13}$C and N measurements were performed on a Delta$^{\text{Plus}}$ XP mass spectrometer
(Thermo Scientific, Bremen, Germany) coupled with an automated elemental
analyzer (Flash EA1112, CE Instruments, Wigan, UK) in a continuous flow mode, at
the Stable Isotope Laboratory of the College of Resources and Environmental
Sciences, China Agricultural University. The carbon isotopic ratios were reported in
the delta notation relative to the V-PDB standard. For this measurement, we obtained
standard deviations low than 0.15‰ for $\delta^{13}$C among replicate measurements of the
same sample. And standard deviations for the N measurements were 0.1%.
The chlorophyll contents of all samples were also determined. The samples were
first extracted by 95% ethyl alcohol (0.5 g sample to 25 mL ethyl alcohol), and then
the absorbancy was measured under the wave length of 665 and 649 mm by the
spectrophotometer. The content of chlorophyll a, b was calculated by the follow
equations:





Chlorophyll a (mg/L) = 13.95×OD665-6.88×OD649                    (6)

Chlorophyll b (mg/L) = 24.96×OD649-7.32×OD665                    (7)

where OD665 and OD649 are the absorbancy under the wave length of 665 and
649mm, respectively.
**2.6 Calculation of the degree of bundle-sheath leakiness**
The degree of bundle-sheath leakiness (φ) was calculated by the transformation of Eq.

(2):

$$\varphi = \left( \frac{(\delta^{13}C_{air} - \delta^{13}C_{plant})/(1+\delta^{13}C_{plant}/1000) - a}{c_i/c_a} + a - b_4 \right) \Big/ (b - s)$$                    (8)

In this equation, parameters a, $b_4$, b and s are constant, while $\delta^{13}C_{plant}$ and $c_i/c_a$ are

the measured values of our samples. We did not measure the $\delta^{13}C_{air}$ at our study site,
so we had to use an approximation of the $\delta^{13}C_{air}$ to do this φ calculation. The
approximated value we used is -9.77‰, which has been measured at Donglingshan
Mountain, Beijing, north China in September 2019. We believe that the two sites
should have similar $\delta^{13}C_{air}$ because the two sites are located in countryside with less
human activities and have a similar distance from the nearest city. The straight line
distances between Donglingshan Mountain and the city center of Beijing, and
between our study and Urumqi city about 90 km.
**2.7 Statistical analysis**
Statistical analyses were conducted using SPSS software (SPSS for Windows, Version
20.0, Chicago, IL, United States). One-way analysis of variance (ANOVA) and
two-way analysis of variance (ANOVA) were used to compare the difference of $\delta^{13}C$
and other physiological traits between each treatment. Pearson analysis was used to





determine the correlation among $\delta^{13}C$, WUE and $c_i/c_a$ in *H. ammodendron*.

## 3 Results

### 3.1 Plant $\delta^{13}C$ under water and nitrogen addition

The $\delta^{13}C$ of the assimilating branches of *H. ammodendron* in the six treatments
W0N0, W0N1, W0N2, W1N0, W1N1, W1N2 was -14.18 ±0.19 ‰, -14.71 ±0.35 ‰,
-14.45 ±0.18 ‰, -14.67 ±0.40 ‰, -14.65 ±0.38 ‰, -14.34 ±0.29 ‰, respectively.
One-way ANOVA analyses showed no significant variation in $\delta^{13}C$ across treatments
(p = 0.788, Fig. 1). Two-way ANOVA analyses suggested that $\delta^{13}C$ was not affected
by water addition (p = 0.678), N addition (p = 0.607) and their interaction (p = 0.563,
Table 1).

Fig.1

Table 1

### 3.2 Gas exchange and WUE under water and nitrogen addition

Photosynthetic rate (A), stomatal conductance ($g_s$), transpiration rate (E) and $c_i/c_a$
ranged from 12.11 μmol $CO_2$ m$^{-2}$ s$^{-1}$ to 39.35 μmol $CO_2$ m$^{-2}$ s$^{-1}$, from 0.09 mol $H_2O$
m$^{-2}$ s$^{-1}$ to 0.31 mol $H_2O$ m$^{-2}$ s$^{-1}$, from 2.87 mmol $H_2O$ m$^{-2}$ s$^{-1}$ to 8.49 mmol $H_2O$ m$^{-2}$ s$^{-1}$
and 0.11 to 0.57, respectively. One-way ANOVA analyses showed significant changes
in leaf gas exchange across the six treatments (p = 0.012 for A, p = 0.006 for $g_s$, p =
0.002 for E and $c_i/c_a$, Fig. 2). Two-way ANOVA analyses suggested that water
addition had exerted effect on $c_i/c_a$ (p = 0.004), that N additions influenced A (p =
0.008) and $c_i/c_a$ (p = 0.009), and that the interaction between water and N supply





played a role in $g_s$ ($p < 0.001$), E ($p < 0.001$) and $c_i/c_a$ ($p < 0.001$, Table 1).
Fig. 2
Instantaneous WUE (ins-WUE) and intrinsic WUE (int-WUE) ranged from 3.09
µmol $CO_2$ / mmol $H_2O$ to 8.49µmol $CO_2$ / mmol $H_2O$ and from 93.64µmol $CO_2$ / mol
$H_2O$ to 208.47µmol $CO_2$ / mmol $H_2O$, respectively. One-way ANOVA analyses
showed significant changes in these two indexes (both $p < 0.001$, Fig. 3). Two-way
ANOVA analyses suggested that water addition, N addition and their interaction all
have significant effect on these two indexes (all $p < 0.05$, Table 1).
Fig. 3
**3.3 Correlations among $\delta^{13}C$, WUE and $c_i/c_a$ ratio**
In order to test whether $\delta^{13}C$ in *H. ammodendron* can indicate WUE, the relationships
among $\delta^{13}C$, ins-WUE, int-WUE and $c_i/c_a$ ratio were revealed in this study. Our
results showed no correlation between $\delta^{13}C$ and ins-WUE ($p = 0.229$, Fig. 4a),
between $\delta^{13}C$ and int-WUE ($p = 0.229$, Fig. 4c), and between $\delta^{13}C$ and $c_i/c_a$ ratio ($p =$
0.183, Fig. 4e). However, there was a negative correlation between ins-WUE and $c_i/c_a$
ratio ($p < 0.001$, Fig. 4b), and between int-WUE and $c_i/c_a$ ratio ($p < 0.001$, Fig. 4d).
Fig. 4
**3.4 The degree of bundle-sheath leakiness under water and nitrogen addition**
The calculated φ ranged from 0.32 to 0.59 with a mean value of 0.45. One-way
ANOVA analyses showed no significant variation in φ across treatments ($p = 0.768$,
Fig. 5). Two-way ANOVA analyses suggested that $\delta^{13}C$ was not affected by water
addition ($p = 0.644$), N addition ($p = 0.600$) and their interaction ($p = 0.521$, Table 1).



Fig. 5

## 4 Discussion

The $\delta^{13}C$ of the assimilating branches in *H. ammodendron* did not change across
treatments (Fig. 1, Table 1), suggesting that neither water addition nor nitrogen
addition influenced the $\delta^{13}C$ of *H. ammodendron*. Previous studies also reported no
significant relationship between $\delta^{13}C$ of $C_4$ plant and water availability (Swap et al.,
2004; Wang et al., 2008), and between $\delta^{13}C$ of $C_4$ plant and nitrogen availability (Yao
et al., 2011, Yang et al., 2017).

In general, the effects of water availability and nitrogen availability on $\delta^{13}C$ are

dependent on $c_i/c_a$ ratio, which reflects the balance between stomatal conductance ($g_s$)
and photosynthetic rate (A) (Farquhar and Richards, 1984). With more water
availability under water addition, plants tend to open stomata to absorb more $CO_2$,
leading to an increase in $g_s$. Two-way ANOVA analyses suggested that water addition
had no effect on both A and $g_s$ (Table 1). However, One-way ANOVA analyses
showed that $g_s$ was higher in W1N0 than that in W0N0 (Fig. 2b). Thus, at least water
addition had a positive effect on $g_s$ under ambient N condition. Increasing $g_s$ under
water supply will lead to the rise of intercellular $CO_2$ because of the decrease of
diffusional resistance to $CO_2$. As the results, $c_i/c_a$ ratio was observed to increase with
increasing moisture (Fig. 2d, Table 1). However, $\delta^{13}C$ remained stable under water
addition (Fig. 1, Table 1). Thus, $c_i/c_a$ ratio could not explain the observed response of
$\delta^{13}C$ to water supply.





For most plants in natural ecosystems, nitrogen is the key factor limiting plant
growth (Hall et al., 2011). Thus, nitrogen addition usually causes plants to absorb
more N. However, extreme drought could prevent plants from absorbing N even
under high N supply. In the present experiment, N supply was found to have an effect
on N contents in *H. ammodendron*. Relative to the control treatment (W0N0), N
contents increased with N supply under low N addition, but kept unchanged under
high addition (Table S1, S2). Nitrogen is the main constituent of Rubisco (ribulose‐
1,5‐bisphosphate carboxylase oxygenase) and chlorophyll in plants. Thus,
chlorophyll a was found to have the similar pattern as N contents under water and N
supply. Chlorophyll a was higher in W0N1 than W0N0, and there was no difference in
chlorophyll a between W0N0 and W0N2 (Table S1). Increasing chlorophyll contents
in W0N1 should lead to the increase in photosynthetic rate (A). However, different
from our prediction, one-way ANOVA analyses suggested that A in W0N1 did not
differ from that in W0N0, and that A in W0N2 is lower than that in W0N0 (Fig., 2a).
Two-way ANOVA analyses showed that N addition had an influence on A (Table 1).
Both the analyses suggested that N supply played a negative role in A, and thus the
consumption of intercellular $CO_2$. Consequently, $c_i/c_a$ ratio was found to increase with
N supply (Fig. 2d, Table 1). Therefore, the variations in $c_i/c_a$ ratio with N addition
could not account for the unchanged pattern in $\delta^{13}C$ under N supply (Fig. 1).
The co-application of water and nitrogen was found to have a negative effect on A
but no effect on $g_s$ (W0N0 vs. W1N1, W1N2, Fig. 2a, b). The responses of A and $g_s$ to
the co-application of water and nitrogen resulted in an increase in $c_i/c_a$ ratio (Fig., 2d).





Since $\delta^{13}C$ remained unchanged under the co-application of water and nitrogen (Fig.
1), $c_i/c_a$ ratio could not also explain the observed $\delta^{13}C$ response to the co-application
of water and nitrogen.
Two underlying mechanisms may explain the observed $\delta^{13}C$ stability across
treatments. The first one is associated with the $\varphi$ value in *H. ammodendron*. For $C_4$
plants, the relationship between carbon isotope discrimination ($\Delta$) and $c_i/c_a$ ratio is
dependent on $\varphi$ values (Ellsworth and Cousins, 2016; Ellsworth et al., 2017; Farquhar,
1983; Wang et al., 2008). Some studies suggested that $\varphi$ value was stable for a given
species under a wide range of environmental conditions (Henderson et al., 1992;
Wang et al., 2008; Cernusak et al., 2013). However, other studies had different
conclusions that $\varphi$ value was influenced by irradiation (Bellasio and Griffiths, 2014;
Kromdijk et al., 2010; Pengelly et al., 2010; Ubierna et al., 2013), temperature (von
Caemmerer et al., 2014), water stress (Fravolini et al., 2002; Gong et al., 2017;
Williams et al., 2001; Yang et al., 2017) and nitrogen supply (Fravolini et al., 2002;
Meinzer and Zhu, 1998; Yang et al., 2017). In current study, the $\varphi$ value of *H.*
*ammodendron* remained unchanged across six treatments (Fig. 5), and two-way
ANOVA analyses suggested that water supply and N supply had no effect on $\varphi$ (Table
1). Therefore, the $\varphi$ value of *H. ammodendron* was insensitive to water and N addition
in this study. Even if the $\varphi$ value remains stable, the relationship between $\Delta$ and $c_i/c_a$
ratio is also associated with the magnitude of the $\varphi$ value. Cernusak et al. (2013)
predicted that when $\varphi$ value is greater than 0.37, the correlation between $\Delta$ and $c_i/c_a$
ratio is positive; conversely, when $\varphi$ value is less than 0.37, the correlation is negative.



In particular, when φ value is equal to 0.37, no significant correlation can be found,
because the coefficient ($[b_4 + φ (b - s) − a]$ in Eq. (2)) of $c_i/c_a$ ratio equals to 0
(Cernusak et al., 2013). The φ value ranged from 0.32 to 0.59 with a mean value of
0.45 in present study, thus the correlation between $Δ$ and $c_i/c_a$ in *H. ammodendron*
should be positive based on the prediction by Cernusak et al. (2013). However, $δ^{13}C$
was found to have no correlation with the measured $c_i/c_a$ ratio (Fig. 4e), suggesting
that the φ value of *H. ammodendron* could be close to 0.37. The reason resulting in
the inconsistence between our calculated φ value and the φ value based on the
prediction by Cernusak et al. (2013) is that we took the atmospheric $δ^{13}C$ data at
Donglingshan, Beijing as $δ^{13}C_{air}$ to calculate φ value. Since atmospheric $δ^{13}C$ is
characterized by geography, the calculation might overestimate the φ value. Therefore,
considering that no correlation was found between with $δ^{13}C$ and $c_i/c_a$ ratio, we
hypothesize that the φ value of *H. ammodendron* could be close to 0.37, which leaded
to the observed insensitive response of $δ^{13}C$ to water and N addition.
The second mechanism is associated with carbonic anhydrase (CA) in $C_4$ plants.
Cousins et al. (2006) suggested that enzymatic activity of CA affects carbon isotope
discrimination in most $C_4$ plants because CA can result in the changing of parameter
$b_4$ (see Eq. (2)). But in traditional view, the parameter $b_4$ was a constant. However, it
is only true when the ratio of PEP carboxylation rate to the $CO_2$ hydration rate ($V_p/V_h$)
is equal to zero, which is caused by a high CA activity. If $V_p/V_h$ is not zero, $b_4$ will
change and be controlled by $V_p/V_h$ (Cousins et al., 2006). Previous studies reported
that CA activity is low in most $C_4$ plants (Cousins et al., 2006; Gillon and Yakir, 2000,





2001; Hatch and Burnell, 1990). Thus, CA activity in *H. ammodendron* might also be
low, leading to the change in $b_4$ with $V_p/V_h$, and thus $\delta^{13}C$. Cousins et al. (2006)
added $V_p/V_h$ into the discrimination pattern of $C_4$ plants and predicted that at a given
$\varphi$ value, when the $V_p/V_h$ is 0 or 1, the correlation between $\Delta$ and $c_i/c_a$ ratio is negative
or positive, respectively. Since CA activity is low in most $C_4$ plants, and the $V_p/V_h$
always ranges from 0 to 1, we speculate that no correlation between $\Delta$ and $c_i/c_a$ ratio
may also occur when the $V_p/V_h$ is a certain value between 0 and 1. Thus, the
uncorrelated pattern between $\Delta$ and $c_i/c_a$ ratio in *H. ammodendron* might be related to
this specific $V_p/V_h$ value due to low CA activity.

Henderson et al. (1992) found that $\delta^{13}C$ of 10 $C_4$ species has negative correlation

with their WUE, which was just opposite to a positive relationship between $\delta^{13}C$ and
WUE for $C_3$ plants. The underlying mechanism of the negative correlation between
$\delta^{13}C$ and WUE is that the $\varphi$ values in 10 $C_4$ species was observed to remain around
0.21 over a range of irradiance and leaf temperature. According to the suggestion by
Cernusak et al. (2013) that $\Delta$ is negatively related to $c_i/c_a$ ratio when $\varphi$ value is less
than 0.37, thus, the $\delta^{13}C$ of 10 $C_4$ species has a positive correlation with $c_i/c_a$ ratio. In
general, under fixed ambient $CO_2$ concentration, WUE is always negatively correlated
with $c_i/c_a$ ratio (see Eq. (3) and Eq. (5)). As a result, a negative relationship between
$\delta^{13}C$ and WUE was observed for the 10 $C_4$ species. However, our study shows that
$\delta^{13}C$ remained stable under water and nitrogen addition (Fig. 1, Table 1), while the
measured ins-WUE and int-WUE was higher in the control treatment (W0N0) than
other treatments (Fig. 3), suggesting and water and N supply had a significant effect





on WUE (Table 1). Furthermore, ins-WUE and int-WUE both had no correlation with
$\delta^{13}C$ (Fig. 4a, 4c). Thus, $\delta^{13}C$ of *H. ammodendron* could not indicate its WUE. The
probable cause of no correlation between WUE and $\delta^{13}C$ is that no correlation has
been found between with $\delta^{13}C$ and the measured $c_i/c_a$ ratio (Fig. 4e), because $c_i/c_a$
ratio is the link between WUE and $\delta^{13}C$.

**5 Conclusion**
Global changes including precipitation and atmospheric N deposition have been
proved to have an important influence on ecosystems, especially for the arid
ecosystems. The present study showed that water and N addition had little effect on
the $\delta^{13}C$ values of *H. ammodendron*, but played an important role in the change of its
gas exchange and water use efficiency (WUE). In addition, no correlation between
instantaneous WUE (ins-WUE) and $\delta^{13}C$, and between intrinsic WUE (int-WUE) and
$\delta^{13}C$ has been found in this study, suggesting that $\delta^{13}C$ of *H. ammodendron* could not
indicate its WUE. This result is caused by the lack of the correlation between $\delta^{13}C$ and
the ratio of intercellular to ambient $CO_2$ concentration ($c_i/c_a$), which might be
associated with the degree of bundle-sheath leakiness ($\varphi$) or the activity of carbonic
anhydrase (CA). Thus, the current experiment implies that the availability of $\delta^{13}C$ as
the indicator of WUE could be not universal for $C_4$ species.

**Conflict of interest**
None declared.




**Funding**

This research was supported by the Chinese National Basic Research Program (No.
2014CB954202 and a grant from the National Natural Science Foundation of China
(No. 41772171).

**Authors' Contributions**

G Wang and J Li designed the experiment and modified the manuscripts. Z Chen
designed and executed the experiment and wrote the manuscripts. X Liu designed the
experiment. X Cui executed the experiment. Y Han executed the experiment.

**Acknowledgements**

This research was supported by the Chinese National Basic Research Program (No.
2014CB954202 and a grant from the National Natural Science Foundation of China
(No. 41772171). We would like to thank the supports from the Fukang Observation
Station of Desert Ecology, Xinjiang Institute of Ecology and Geography, Chinese
Academy of Sciences, and to thank Ma Yan for analyzing stable carbon isotope ratios
in the Isotope Lab at the College of Resources and Environment, China Agricultural
University.

**Data availability**

The datasets analyzed in this manuscript are not publicly available. Requests to access





the datasets should be directed to gawang@cau.edu.cn.

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

Table 1 The p values of all measured and calculated indexs in plants under two-way ANOVA
analysis of water (W) and nitrogen (N) additions

|  | W | N | W*N |
|---|---|---|---|
| $\delta^{13}C$ | 0.678 | 0.607 | 0.563 |
| Photosynthetic rate (A) | 0.331 | 0.008** | 0.183 |
| Stomatal conductance ($g_s$) | 0.533 | 0.871 | <0.001*** |
| Transpiration rate (E) | 0.622 | 0.883 | <0.001*** |
| $c_i/c_a$ | 0.004** | 0.009** | <0.001*** |
| ins-WUE | 0.002** | <0.001*** | <0.001*** |
| int-WUE | 0.004** | 0.018* | <0.001*** |
| φ | 0.644 | 0.600 | 0.521 |

Note. *, **, *** indicates a significant influence.













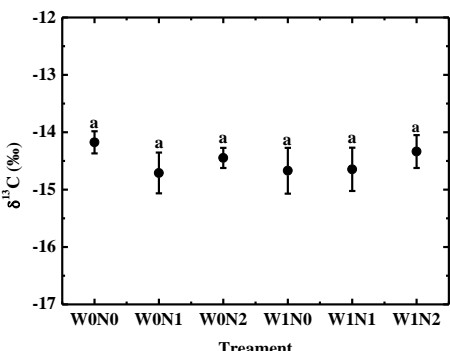


Fig. 1 The $\delta^{13}C$ of assimilating branches of *Haloxylon ammodendron* under water (W) and

nitrogen (N) additions. The spot represents the mean value of four replicates with error bars

denoting the standard error (SE).


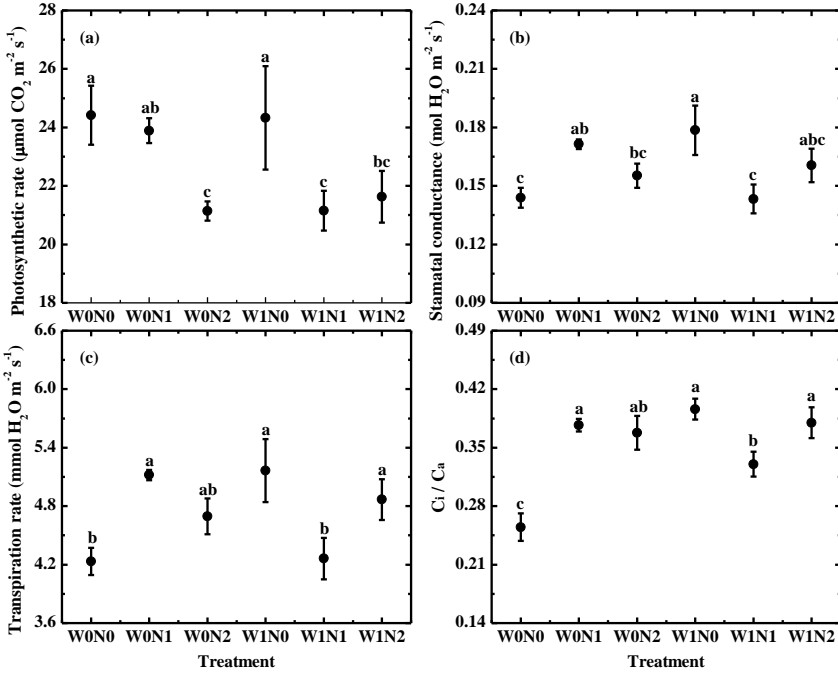


Fig. 2 Variations in photosynthetic rate (a), stomatal conductance (b), water use-efficiency (c) and

$c_i/c_a$ (d) across water (W) and nitrogen (N) additions. The spot represents the mean value of four

replicates with error bars denoting the standard error (SE).





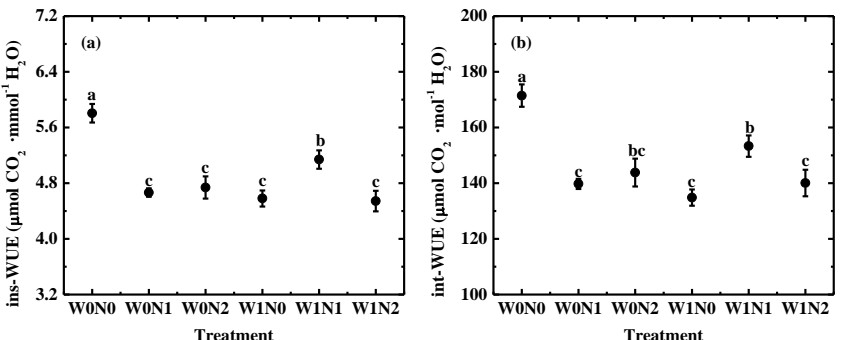

Fig. 3 Variations in ins-WUE (a) and int-WUE (b) across water (W) and nitrogen (N) additions. The spot represents the mean value of four replicates with error bars denoting the standard error (SE).





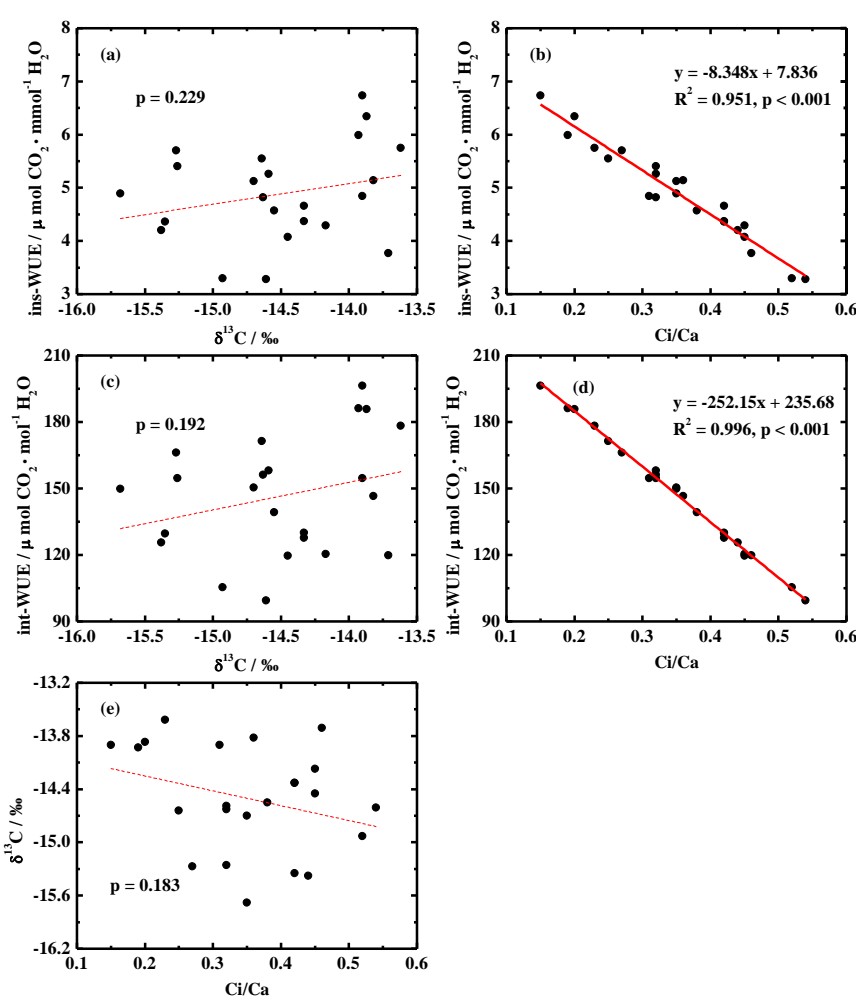


Fig. 4 Correlations of ins-WUE vs. $\delta^{13}$C (a), ins-WUE vs. $c_i/c_a$ (b), int-WUE vs. $\delta^{13}$C (c),
int-WUE vs. $c_i/c_a$ (d), and $\delta^{13}$C vs. $c_i/c_a$ (e) of assimilating branches of *Haloxylon ammodendron*





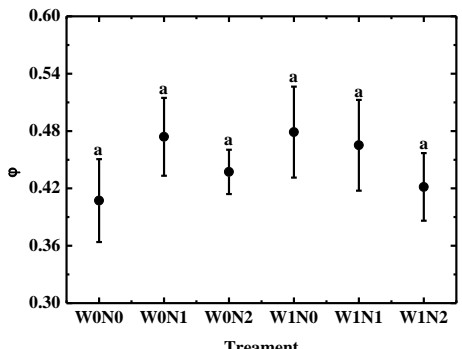


Fig. 5 Variations in φ across water (W) and nitrogen (N) additions. The spot represents the mean
value of four replicates with error bars denoting the standard error (SE).
