# Peer review of "Evaluating the response of $\delta^{13}\text{C}$ in *Haloxylon ammodendron*, a"

_Biogeosciences, 2020_

## Referee Comment (RC1) · Anonymous Referee #1 · 4 Nov 2020

Water use efficiency is an important parameter reflects plant's adaptation to changes in soil water availability. This study assessed whether or not there are linkage between leaf carbon isotope composition and water use efficiency in a C4 shrub, Haloxylon ammodendron growing under different water and nitrogen conditions. The authors reported that leaf photosynthetic parameters, but not carbon isotope composition, were affected by water and nitrogen addition treatments. However, carbon isotope composition of assimilating branch was not correlated with water use efficiency, which may have

resulted from bundle sheath leakiness and lower activity of carbonic anhydrase. The topic of the study is very interesting and the results are useful for the prediction of plant drought adaptation. However, lack of correlation between carbon isotope composition of assimilating branch and water use efficiency may have also resulted from differences in temporal scale. Carbon isotope composition was determined by the growing condition and photosynthetic discrimination of the period of formation (days); whereas WUE was calculated based on short-term gas exchange measurements (mins). In another word, the measured branch may have emerged a few weeks or even months before the leaf gas exchange measurements. Lack of treatment effects on carbon isotope composition of assimilating branch may associate with rooting depth and water sources. The studied C4 shrub has the ability to access and uptake groundwater, which will reduce its dependency on precipitation and water addition treatment. The authors need to provide more information on plant water sources. Meteorological information is required for the experimental year. L189, using of 450 mmol mol-1 CO2 need to be justified. L191, why the authors used 1000 umol m-2 s-1, but not higher photon flux density, such as 1500 or 2000? L191, what are the meanings of "kept stable"? This sentence need to be described more clearer.

---

## Referee Comment (RC2) · Anonymous Referee #2 · 17 Dec 2020

Authors have investigated the effect of precipitation change and increasing atmospheric N deposition on $\delta$13C of H. ammodendron. Numerous studies have doubted the ability of the carbon isotopic composition of C4 plants as a tracer of water-use efficiency. The existing observations on positive/negative correlation between $\delta$13C of C4 plants and precipitation change are extremely scanty, and most of the global records advocate for neutral relationship between them. I have some serious doubts on the experimental procedures which are as follows: (I) the age of the sampled leaves have not

[Figure]

been mentioned in the manuscript. Usually, plants attain the final $\delta$13C at the end of the growing season, and there is a considerable difference between buds, young and matured leaves. The time-lag between actual $\delta$13C fixation in the plant and duration of the experimental process could be a valid reason for the correlation obtained in this work. (II) some of the experimental results do not give any definite conclusion. For example, authors have calculated the leakiness values, but at later stage discarded the values citing the no correlation between $\delta$13C and WUE. Rather than simply rejecting their measured leakiness values, authors should provide alternative concepts. Otherwise, the whole exercise becomes futile and should not be incorporated in the main text. Furthermore, desert environments are home for extreme climatic conditions; high day-time temperature, very low precipitation, strong difference between day and night time temperatures. Therefore, the $\delta$13C value of modern desert plants cannot be used an analogue of all the C4 plants, and this work has very limited application. In terms of the presentation, the manuscript is poorly organised, difficult to follow at places, includes wrong information (example line 93) and does not provide any new insight into the subject. As per my opinion, the manuscript, at its current form, does not meet the standard for the publication in Biogeosciences. As authors have conducted tedious experiments, I am open to review the revised manuscript if they substantially modify the manuscript after addressing all the afore-mentioned points.

---

## Author Comment (AC1) · 5 Jan 2021

Thank you for your comments! Indeed, we also recognized that differences in temporal scale may lead the carbon isotope composition ($\delta$13C) to be irrelevant to water use efficiency (WUE). Yet we think our conclusion should be believable, because both water addition and N addition changed the WUE of H. ammodendron, but $\delta$13C did not show variability across water and N treatments. We will discuss this uncertainty in

subsequent revisions. We agreed with your comments that utilization of groundwater leads to the insensitivity of H. ammodendron to rainfall and water addition. In fact, we have found that the root of H. ammodendron can be inserted into the soil layer deeper than 3 m, which makes it easier to absorb groundwater. Thus, this is one of the mechanisms behind our observed results. We will try our best to supplement plant water source information and meteorological information in subsequent revised editions, and discuss their influence on our results. Thank you very much for your suggestion. The photo flux density is 1600 mmol m-2 s-1 in our measurement. We considered that it is more suitable for measuring gas exchange in H. ammodendron, which grew up in a desert area with high light intensity. The 1000 mmol m-2 s-1 written in the previous version is our clerical error. Thank you for picking out our errors. The meaning of "kept stable" is that leaf temperatures were within the range of 29.5 °C to 30.5 °C during the entire period of gas exchange measurements. We will check our manuscript carefully and improve our writing in subsequent revisions.

---

## Author Comment (AC2) · 5 Jan 2021

Thank you for your comments! Leaf is most effective for the assessment of plant carbon isotope composition ($\delta$13C). However, the leaves of H. ammodendron are degenerated due to extreme drought, we had to collect the assimilating branches of H. ammodendron, which was its prime assimilating organ. We know that there is a considerable difference in $\delta$13C between buds, young and matured leaves, so we collected the ma-

tured assimilation branches at the top of the treetops in each individual, which were synthesized in the year of sampling, to minimize the effect of the age on $\delta$13C. We will provide more information about the sampling in subsequent revised editions. In addition, we also recognized that differences in temporal scale may lead the $\delta$13C to be irrelevant to water use efficiency (WUE). Yet we think our conclusion should be believable, because both water addition and N addition changed the WUE of H. ammodendron, but $\delta$13C did not show variability across water and N treatments. We will discuss this uncertainty in subsequent revisions. Thanks for your suggestion on the $\varphi$ value, The definite conclusion obtained from this study is that the $\varphi$ value of H. ammodendron does not change with water and N addition, implying that environmental conditions may have no influence on $\varphi$ value, which is still inconclusive. We will add this conclusion in subsequent revisions. The results gained from the present study are not necessarily analogous to all C4 plants due to the extreme climatic conditions in desert ecosystem. Yet we believed that this work has important application for enhancing our understanding of physiological responses of desert plants to future changes in precipitation and atmospheric N deposition. This is because H. ammodendron is a dominant species in desert regions, especially in Asia desert, which has a great effect on the stabilization of sand dunes, the survival and development of understory plants and the structure and function of desert ecosystems (Sheng et al., 2005; Su et al., 2007; Cui et al., 2017). Thus, the prediction of plant drought adaptation in H. ammodendron is crucial in desert ecosystem. We will supplement the significance of this study in subsequent revised editions. Thanks for your suggestion!

---

## Author Response (AR1)

**Reviewer 1**

Water use efficiency is an important parameter reflects plant's adaptation to changes in soil water availability. This study assessed whether or not there are linkage between leaf carbon isotope composition and water use efficiency in a C4 shrub, Haloxylon ammodendron growing under different water and nitrogen conditions. The authors reported that leaf photosynthetic parameters, but not carbon isotope composition, were affected by water and nitrogen addition treatments. However, carbon isotope composition of assimilating branch was not correlated with water use efficiency, which may have resulted from bundle sheath leakiness and lower activity of carbonic anhydrase. The topic of the study is very interesting and the results are useful for the prediction of plant drought adaptation. However, lack of correlation between carbon isotope composition of assimilating branch and water use efficiency may have also resulted from differences in temporal scale. Carbon isotope composition was determined by the growing condition and photosynthetic discrimination of the period of formation (days); whereas WUE was calculated based on short-term gas exchange measurements (mins). In another word, the measured branch may have emerged a few weeks or even months before the leaf gas exchange measurements. Lack of treatment effects on carbon isotope composition of assimilating branch may associate with rooting depth and water sources. The studied C4 shrub has the ability to access and uptake groundwater, which will reduce its dependency on precipitation and water addition treatment. The authors need to provide more information on plant water sources. Meteorological information is required for the experimental year. L189, using of 450 mmol mol-1 CO2 need to be justified. L191, why the authors used 1000 umol m-2 s-1, but not higher photon flux density, such as 1500 or 2000? L191, what are the meanings of "kept stable"? This sentence need to be described more clearer.

**Response:**

Thank you for your comments! Indeed, we also recognized that differences in temporal scale may lead the carbon isotope composition ($\delta^{13}C$) to be irrelevant to water use efficiency (WUE). Yet we think our conclusion should be believable, because both water addition and N addition changed the WUE of *H. ammodendron*, but $\delta^{13}C$ did not show variability across water and N treatments. We have revised the discussion, please see line 401-420.

We agreed with your comments that utilization of groundwater leads to the insensitivity of *H. ammodendron* to rainfall and water addition. In fact, we have found that the root of *H. ammodendron* can be inserted into the soil layer deeper than 3 m, which makes it easier to absorb groundwater. Thus, this is one of the mechanisms behind our observed results. However, the shallow soil water (0-40 cm) and

groundwater are two important water sources for *H. ammodendron* (Dai et al., 2014), and water addition resulted in an increase of soil water contents in shallow soil layer. Moreover, gas exchange changed across treatments in the present study (Fig. 2). Thus, we believed that the utilization of groundwater by *H. ammodendro*n may be one of the reasons why its $\delta^{13}$C is not sensitive to water and N addition, but it should not be the main reason. The discussion above has been added, please see line 380-390. In addition, the meteorological information in the sampling year have been added, please see line 147-149.

The photo flux density is 1600 mmol·m$^{-2}$·s$^{-1}$ in our measurement. We considered that it is more suitable for measuring gas exchange in *H. ammodendron*, which grew up in a desert area with high light intensity. The 1000 mmol·m$^{-2}$·s$^{-1}$ written in the previous version is our clerical error. Thank you for picking out our errors. The meaning of "kept stable" is that leaf temperatures were within the range of 29.5 °C to 30.5 °C during the entire period of gas exchange measurements. We have revised these error, please see 190-194.

**Reviewer 2**

Authors have investigated the effect of precipitation change and increasing atmospheric N deposition on δ13C of H. ammodendron. Numerous studies have doubted the ability of the carbon isotopic composition of C4 plants as a tracer of water-use efficiency. The existing observations on positive/negative correlation between δ13C of C4 plants and precipitation change are extremely scanty, and most of the global records advocate for neutral relationship between them. I have some serious doubts on the experimental procedures which are as follows: (I) the age of the sampled leaves have not been mentioned in the manuscript. Usually, plants attain the final δ13C at the end of the growing season, and there is a considerable difference between buds, young and matured leaves. The time-lag between actual δ13C fixation in the plant and duration of the experimental process could be a valid reason for the correlation obtained in this work. (II) some of the experimental results do not give any definite conclusion. For example, authors have calculated the leakiness values, but at

later stage discarded the values citing the no correlation between δ13C and WUE. Rather than simply rejecting their measured leakiness values, authors should provide alternative concepts. Otherwise, the whole exercise becomes futile and should not be incorporated in the main text. Furthermore, desert environments are home for extreme climatic conditions; high day-time temperature, very low precipitation, strong difference between day and night time temperatures. Therefore, the δ13C value of modern desert plants cannot be used an analogue of all the C4 plants, and this work has very limited application. In terms of the presentation, the manuscript is poorly organised, difficult to follow at places, includes wrong information (example line 93) and does not provide any new insight into the subject. As per my opinion, the manuscript, at its current form, does not meet the standard for the publication in Biogeosciences. As authors have conducted tedious experiments, I am open to review the revised manuscript if they substantially modify the manuscript after addressing all the afore-mentioned points.

**Response:**

Thank you for your comments! Leaf is most effective for the assessment of plant $\delta^{13}C$. However, the leaves of *H. ammodendron are* degenerated due to extreme drought, we had to collect the assimilating branches of *H. ammodendron*, which was its prime assimilating organ. We know that there is a considerable difference in carbon isotope composition ($\delta^{13}C$) between buds, young and matured leaves, so we collected the matured assimilation branches at the top of the treetops in each individual, which were synthesized in the year of sampling, to minimize the effect of the age on $\delta^{13}C$. We have added the information about the sampling, please see line 196-198.

In addition, we also recognized that differences in temporal scale may lead the $\delta^{13}C$ to be irrelevant to water use efficiency (WUE). Yet we think our conclusion should be believable, because both water addition and N addition changed the WUE of *H. ammodendron*, but $\delta^{13}C$ did not show variability across water and N treatments. We We have revised the discussion, please see line 401-420.

Thanks for your suggestion on the φ value, The definite conclusion obtained from

this study is that the φ value of *H. ammodendron* does not change with water and N addition, implying that environmental conditions may have no influence on φ value, which is still inconclusive. We have added this conclusion, please see line 435. In addition, since we found that the mean φ value is 0.45, it seems that φ is not the driver of the observed $\delta^{13}C$ pattern in *H. ammodendron*. However, due to the time-lag of measured $\delta^{13}C$ and $c_i/c_a$, there were some uncertainties in the calculation of φ value based on the measured $\delta^{13}C$ and $c_i/c_a$. Therefore, we considered that the special φ value (0.37) may be one of the mechanisms behind the unchanged $\delta^{13}C$ in *H. ammodendron*. We have revised the discussion about the φ value, please see line 356-362.

The results gained from the present study are not necessarily analogous to all $C_4$ plants due to the extreme climatic conditions in desert ecosystem. Yet we believed that this work has important application for enhancing our understanding of physiological responses of desert plants to future changes in precipitation and atmospheric N deposition. This is because *H. ammodendron* is a dominant species in desert regions, especially in Asia desert, which has a great effect on the stabilization of sand dunes, the survival and development of understory plants and the structure and function of desert ecosystems (Sheng et al., 2005; Su et al., 2007; Cui et al., 2017). Thus, the prediction of plant drought adaptation in *H. ammodendron* is crucial in desert ecosystem. We have added the significance of this study, please see line 421-429.

---

## Author Response (AR2)

1. The author should clarify how long the samples were collected after water and nitrogen addition, as this will affect the results of the measured photosynthetic rate, stomatal conductance and water use-efficiency. More information on this should be added.

**Answer:**

Thank you for your suggestion! I have added the information (please see 192-194, lines 203, lines 223-224 in the marked-up manuscript version). In order to avoid the instantaneous effect of water and nitrogen addition on plant gas exchange, gas exchange measurement was performed before water and nitrogen addition. The measurement of gas exchange was conducted in 27-29, June, while the addition of water and nitrogen were applied in equal amounts, twelve times, once a week in April, July and September (please see lines 192-194, lines 203 in the marked-up manuscript version). The sample collection was carried out on July 20, during the process of adding water and nitrogen in July. This is because the main measurement index of the collected samples is carbon isotope, which is less instantaneously affected by the water and nitrogen addition. We have supplemented the time of sample collection in the revised version, please see lines 223-224 in the marked-up manuscript version.

2. Since the authors chosen H. ammodendron as their study object, the species, habits, characteristics of H. ammodendron should be elaborately described in the manuscript. I suggest the authors add this part of content.

**Answer:**

Your suggestion is correct. Thanks! We have added the information. *H. ammodendron* is a species of Chenopodiaceae, which is a xerophytic and halophytic woody plant (Cui et al., 2017). The leaves of *H. ammodendron* have been completely degraded due to the extreme drought, and the assimilation branches, which are the glossy green branches (Fig. S1), perform the same functions as the leaves. Due to its drought tolerance, H. ammodendron is widely distributed in desert areas. Please see lines 171-177 in the marked-up manuscript version.

3. What is the type of nitrogen used as the fertilizer to simulate the N adding experiment? And whether this type of nitrogen could be effectively used by H. ammodendron.

**Answer:**

Thank you very much! The type of nitrogen used in the N addition is $NH_4NO_3$, Cui et al. (2017) has confirmed that this type of nitrogen could be effectively used by *H. ammodendron*. We have supplemented the type of nitrogen in the revised version, please see lines 191-192 in the marked-up manuscript version.

4. How did you measure the CO2 concentration in the assimilation branches? The assimilation branch is what the part of H. ammodendron. Could the authors show a

picture of the assimilation branches of H. ammodendron in the supplemental materials?

**Answer:**

Sorry, our unclear expression may mislead you. In fact, we did not measure the $CO_2$ concentration in the assimilation branches. However, we obtained ambient $CO_2$ concentration($c_a$) and intercellular $CO_2$ concentration ($c_i$) using LI-6400 portable photosynthesis system. Assimilation branches refer to the green branches of *H. ammodendron*. Due to the deterioration of the leaves, *H. ammodendron* has no leaves, only branches, and uses green branches as assimilation organs (please see lines 174-176 in the marked-up manuscript version). A picture of assimilation branches has been supplemented in the revised version, please see Fig. S1.

5. Have the authors only measured the gas exchange in June? The gas exchange of plants has a large variation in the day and night as well as the different months. Could the data of June represent all the conditions?

**Answer:**

Your concern is correct. Due to the difficulty of the measurement, we did only measure the gas exchange at the end of June. We also recognize that there are day-to-night differences and monthly differences in the gas exchange of plants. However, from the end of June to July is the most important growth period for *H. ammodendron*. So we believe that it may be the most appropriate to take measurements during this period, and the results of the measurements are therefore more representative. Previous studies have also generally conducted this measurement during growing season (Nyongesah and Wang, 2013; Cui, 2018; Gong et al., 2019). The reason why we chose this time period to measure gas exchange has been added in the revised version, please see lines 202-207 in the marked-up manuscript version.

6. Line 195-203: I do not clearly understand what index you measure at here?

**Answer:**

Sorry for our unclear expression. The indexes include photosynthetic rate (A), stomatal conductance ($g_s$), transpiration rate (E), the ambient $CO_2$ concentration ($c_a$) and the intercellular $CO_2$ concentration ($c_i$). We have added this information in the revised version, please see lines 209-213 in the marked-up manuscript version.

7. Section 2.5, When did you sample the assimilation branches? How long have the assimilation branches been cut? And how long is the growth time of the length of the sample collected?

**Answer:**

The sample collection was carried out on July 20, please see lines 223-224 in the marked-up manuscript version. The length of the sample is 15-20 cm, please see line

228 in the marked-up manuscript version. After the samples were collected, they were immediately divided into two parts randomly and taken back to the laboratory at Fukang Station. In the laboratory, the first part was used to determine the chlorophyll content immediately. The second part was immediately inactivated in a 105 °C oven, and then brought back to Beijing in a ziplock bag. The time interval between sample collection and inactivation is very short. After inactivation, the carbon exchange of the assimilating branches stop, so the isotope composition of the samples will not change anymore. These information have been added in the revised version, please see lines 231-241 in the marked-up manuscript version. We believed that these assimilation branches were synthesized in the year of sampling.

8. Could you add some introduction about the mechanism of how the ratios of ci/ca control the δ13C variations of plants?

**Answer:**
Thanks for your suggestion! The pattern of carbon isotopic discrimination ($\Delta$) in $C_4$ plant has been introduced in the paper, please see equation (2). According to this equation and its transformation (equation (3)), if the coefficient $[b_4 + \varphi (b - s) – a]$ is greater than 0, $\delta^{13}C$ decrease with increasing $c_i/c_a$, if this coefficient is lower than 0, $\delta^{13}C$ increase with increasing $c_i/c_a$. This mechanism has been added in the revised version, please see lines 134-136 in the marked-up manuscript version.

9. The δ13Cair has large diurnal and seasonal variations. In section 2.6, I suggest the authors use the range of δ13Cair to replace the specific value.

**Answer:**
Thanks for your suggestion! We have added the range of $\delta^{13}C_{air}$ in the calculation in the revised version, please see lines 273-279 in the marked-up manuscript version.

10. In the Materials and methods, the authors did not clearly introduce how they measured the stomatal conductance (gs) and photosynthetic rate (A).

**Answer:**
Sorry. The measurements of A and gs were introduced in Section 2.4. Please see lines 213-221 in the marked-up manuscript version.

11. Line 306-308 and Line 311-313: the logic of these two sentences is a little chaotic.

**Answer:**
Sorry. We have revised these two sentences, please see lines 353-354 in the marked-up manuscript version.

12. Line 327-330: could the authors explain that why the traditional theory is useless at here?

**Answer:**

Thanks for the suggestion! The observed results may be caused by the extremely high light intensity at the study site. Due to the high light intensity, photosynthetic rate may not be correlated with chlorophyll contents. We have added this discussion in the revised version, please see lines 379-382 in the marked-up manuscript version.

13. Line 400-401, I did not see the correlation plot between Δ and ci/ca ratio.

**Answer:**

Sorry for our unclear expression. According to equation (2), Δ is equal to $\delta^{13}C_{air}$ minus $\delta^{13}C_{plant}$. Thus, no matter how $\delta^{13}C_{air}$ changes, Δ is negatively related to $\delta^{13}C_{plant}$. We found that $\delta^{13}C_{plant}$ was not related to $c_i/c_a$, suggesting that Δ has no correlation with $c_i/c_a$. We have changed our expression in the revised version, please see lines 417-419, 448-450 in the marked-up manuscript version.

14. Line 416-417: "Although this result was just opposite to a positive relationship between δ13C and WUE for C3 plants". Adding references for this sentence.

**Answer:**

Thanks! The references have been added in the revised version, please see line 467 in the marked-up manuscript version.